# Genotype-Phenotype Correlations in 208 Individuals with Coffin-Siris Syndrome

**DOI:** 10.3390/genes12060937

**Published:** 2021-06-19

**Authors:** Ashley Vasko, Theodore G. Drivas, Samantha A. Schrier Vergano

**Affiliations:** 1Clinical Research Unit, Children’s Hospital of The King’s Daughters, Norfolk, VA 23507, USA; ashley.vasko@chkd.org; 2Division of Human Genetics, The Children’s Hospital of Philadelphia, Philadelphia, PA 19104, USA; Theodore.drivas@pennmedicine.upenn.edu; 3Department of Genetics, The University of Pennsylvania, Philadelphia, PA 19104, USA; 4Division of Medical Genetics and Metabolism, Children’s Hospital of The King’s Daughters, Norfolk, VA 23507, USA; 5Department of Pediatrics, Eastern Virginia Medical School, Norfolk, VA 23507, USA

**Keywords:** Coffin-Siris syndrome, genotype-phenotype, BAF complex

## Abstract

Coffin-Siris syndrome (CSS, MIM 135900) is a multi-system intellectual disability syndrome characterized by classic dysmorphic features, developmental delays, and organ system anomalies. Genes in the BRG1(BRM)-associated factors (BAF, Brahma associated factor) complex have been shown to be causative, including *ARID1A*, *ARID1B*, *ARID2*, *DPF2*, *SMARCA4*, *SMARCB1*, *SMARCC2*, *SMARCE1*, *SOX11*, and *SOX4*. In order to describe more robust genotype-phenotype correlations, we collected data from 208 individuals from the CSS/BAF complex registry with pathogenic variants in seven of these genes. Data were organized into cohorts by affected gene, comparing genotype groups across a number of binary and quantitative phenotypes. We determined that, while numerous phenotypes are seen in individuals with variants in the BAF complex, hypotonia, hypertrichosis, sparse scalp hair, and hypoplasia of the distal phalanx are still some of the most common features. It has been previously proposed that individuals with *ARID*-related variants are thought to have more learning and developmental struggles, and individuals with *SMARC*-related variants, while they also have developmental delay, tend to have more severe organ-related complications. *SOX*-related variants also have developmental differences and organ-related complications but are most associated with neurodevelopmental differences. While these generalizations still overall hold true, we have found that all individuals with BAF-related conditions are at risk of many aspects of the phenotype, and management and surveillance should be broad.

## 1. Introduction

Genes in the BAF (Brahma/BRG1-associated factor) complex are essential in chromatin remodeling. Pathogenic variants in this complex have been associated with a number of conditions, including Coffin-Siris syndrome (CSS, MIM 135900, ORPHA:1465), Nicolaides-Baraitser syndrome (NCBRS, MIM 601358) and other CSS-like conditions (*SOX4*) [1,2]. A range of learning and developmental differences, organ-related anomalies, and variable physical and facial features typically characterize BAFopathies, a term that has been proposed in its description. The most well-known among these syndromes is CSS. CSS is a well-described intellectual disability disorder characterized classically by facial features and organ-system anomalies. Historically, CSS was classified upon recognition of typical phenotype changes, hypoplasia of the distal phalanx, hypertrichosis, and sparse scalp hair. Genes in the BAF complex have been shown to be causative, including *ARID1A*, *ARID1B*, *ARID2*, *DPF2*, *SMARCA4*, *SMARCB1*, *SMARCC2*, *SMARCE1*, *SOX11*, and *SOX4*. Individuals display significant variability in terms of learning and developmental differences, as well as cardiac, renal, brain, and skeletal malformations [3]. Some more common physical manifestations include agenesis of the corpus callosum, variable cardiac defects, feeding difficulties, hypotonia, and vision and hearing anomalies.

A number of genotype-phenotype correlations have been conducted in individuals with molecularly confirmed CSS. Work in this field has incorporated genotype-phenotype associations for individuals with pathogenic variants in *ARID1A*, *ARID1B*, *SMARCA4*, *SMARCB1*, *SMARCE1*, and *SOX11* [3,4,5,6,7,8,9]. Current hypotheses suggest that genes in the BAF complex associate with transcription factors that play a role in neurodevelopment [2,8,10]. In general, more significant developmental delays and fewer anatomic anomalies have been described in individuals with *ARID*-related variants, compared to individuals with *SMARC*-related variants in whom developmental delays are thought to be milder, and more severe organ-related complications are seen. *SOX11* has been associated with more neurodevelopmental complications [8]. The exact etiology behind these genotype-phenotype correlations remains unknown. This manuscript aims to incorporate each genotype in a comparative by-gene analysis. This report examines genotype-phenotype correlations in a large cohort of individuals from the CSS/BAF complex registry in an effort to further delineate any differences that may be of use to both families and clinicians caring for these individuals.

## 2. Materials and Methods

### 2.1. Study Population and Data Collection

The CSS/BAF complex registry is an Institutional Review Board-approved study (IRB #15-03-EX-0058), begun in 2015, which enrolls individuals with a molecularly confirmed diagnosis of a BAF-related condition. The registry is a parent-caregiver completed survey, which also includes the collection of medical records. At the time of this submission, there are 341 participants enrolled in the CSS/BAF-related disorders registry. While most individuals are from the United States, there are registrants from many other countries, including Canada, Australia, the United Kingdom, France, Greece, China, India, and others. The ages of individuals range from infancy to adulthood. Parent-completed surveys and medical records are used to collect clinical data, with information on development, facial features, growth patterns, and medical conditions being examined. Information and secured files are collected via RedCap© data systems.

Individuals were included in our sample population if there was sufficient data supporting their phenotypes. Only individuals for whom we had a confirmed molecular diagnosis of a BAF-related disorder were reviewed; pertinent and available medical records were also examined to validate and expand the data. Of the 341 patients, 208 were found to have enough data for inclusion in our subsequent analyses. We organized the data into cohorts by affected gene, comparing genotype groups across a number of binary and quantitative phenotypes.

To assess developmental delays in each genotype, we utilized the Denver Developmental Screening Test II (DDST-II). The DDST-II is a screening tool used to assess the progression of children who are at risk of developmental delays. The child’s age, in months, is demonstrated by the horizontal axis on the DDST-II. The milestones, of which there are 125, are demonstrated by the vertical axis on the DDST-II. The DDST-II is utilized by determining the child’s chronological age and marking it in the screening tool. Hypothetically, the neurotypical child should have accomplished all milestones to the left of their chronological age. We ran a BAF complex by-gene analysis of five developmental milestones, including walk, sit, first word, roll, and crawl acquisition. We took the age that 50% of children, demonstrated by the DDST-II, accomplished these five milestones. We compared these ages to the average age each genotype completed each milestone. We then calculated the percent delays to determine how delayed each individual was in completing each milestone.

### 2.2. Statistical Analyses

Data describing binary phenotypes were visualized in R, displaying the proportion of patients, divided by genotype group, affected by each phenotype. These data were analyzed using the Chi-square test, dividing individuals by genotype and comparing the count of individuals within each group affected by each phenotype. *p*-values for phenotypes with nominally statistically significant differences across genotype groups (*p* < 0.05) are indicated. For each phenotype with nominally statistically significant differences, genotypes with an absolute normalized Pearson residual greater than 2 are indicated with an asterisk. Quantitative traits were also visualized in R, displaying the mean value per genotype group +/− the standard error of the mean. These data were analyzed by ANOVA with Tukey’s post hoc test to identify significant differences across genotype groups. Nominally statistically significant (*p* < 0.05) differences between gene groups are indicated. Altogether, 36 independent statistical analyses were performed for a study-wise Bonferroni-adjusted *p*-value significance threshold of 0.0014.

## 3. Results

### 3.1. Results

#### 3.1.1. Cohort Description

We identified 208 individuals in our cohort with molecularly confirmed pathogenic variants along the BAF complex and with sufficient phenotypic information for inclusion in our study. Pathogenic variants within ARID1B (*n* = 130, 63%) and SMARCA4 (*n* = 32, 15%) were found to be the most common within our cohort. The remaining 22% of our CSS patients were found to have variants in the following genes (Table 1): ARID1A (*n* = 15, 7%), SMARCB1 (*n* = 14, 7%), ARID2 (*n* = 8, 4%), SOX11 (*n* = 5, 2%), and SMARCE1 (*n* = 4, 2%). Males account for 60% (*n* = 125) of individuals in our sample population, females account for the additional 40% (*n* = 83). Across our entire CSS cohort, the four most common phenotypes are as follows: hypertrichosis (109/208, 52%), sparse scalp hair (98/208, 47%), hypotonia (89/208, 43%), and hypoplasia of the distal phalanx (85/208, 41%).

#### 3.1.2. Phenotype Generalizations 

To determine the frequency of CSS phenotypes within patients affected by pathogenic variants in different BAF complex components, we analyzed 28 classic CSS phenotypes in patients with pathogenic variants in each of the seven BAF complex genes represented in our cohort. The results of these analyses are shown in Table 2. Sparse scalp hair (8/15, 53%), hypoplasia of the distal phalanx (8/15, 53%), and strabismus (8/15, 53%) were the most common phenotypes reported in individuals with variants in ARID1A. In the ARID1B population, hypertrichosis (81/130, 62%) was the only phenotype reported in the majority of individuals. Sparse scalp hair (5/8, 63%) and global developmental delay (4/8, 50%) were seen in a majority of ARID2 patients. Hypoplasia of the distal phalanx was seen in the majority of SMARCA4 patients (20/32, 63%). Cryptorchidism, sparse scalp hair, and hypertrichosis were all seen in 50% of SMARCB1 patients (7/14, 3/6 males for cryptorchidism). The SMARCE1 and SOX11 groups each had five or fewer patients, making it difficult to make any general statements about phenotype frequency in these populations. 

#### 3.1.3. Phenotype Frequency

To assess for differences in phenotype frequency across different CSS patient groups, the frequency of each phenotype across all genotype groups was compared (Figure 1A). Altogether, five phenotypes were identified with nominally statistically significant differences across CSS genotype groups: fifth digit hypoplasia (*p* = 3.185 × 10^−3^), hypertrichosis (*p* = 1.381 × 10^−2^), kidney malformations (*p* = 1.294 × 10^−6^), microcephaly (*p* = 2.694 × 10^−3^), and macrocephaly (*p* = 3.852 × 10^−2^). It should be noted that some of our genotype groups (notably SOX11 and SMARCE1) are composed of very few patients, making the interpretation of some of these statistical results more difficult for these groups in particular. For kidney malformations, these significant differences were primarily driven by the relative overabundance of kidney malformations in individuals with ARID1A (normalized Pearson residual of Chi-square analysis = 2.19), SMARCE1 (residual = 3.44), and SMARCB1 (residual = 3.1) variants, and an underabundance of kidney malformations in individuals with ARID1B (residual = −2.18) variants (Appendix A). The significant differences in microcephaly were chiefly driven by the overabundance of this phenotype in the SMARCA4 patients (residual = 3.3), and the significant differences in macrocephaly were driven by the corresponding underabundance of this phenotype in the SMARCA4 (relative residual = −2.08) group (Appendix A). For fifth digit hypoplasia, there is a trend towards the enrichment of the phenotype in SMARCA4 (residual = 1.91) and SMARCE1 (residual = 1.85) patients, and a trend towards underrepresentation of this phenotype in ARID1B (residual = −1.39) and ARID2 (residual = −1.26) patients. For the hypertrichosis phenotype, there is a trend towards the enrichment of this phenotype in ARID1B patients (residual = 1.48) compared to all other patient groups.

#### 3.1.4. Developmental Differences

Individuals with CSS are known to have a wide range of global developmental differences. To identify patient genotypes with the most significant delays, we ran a BAF complex by-gene analysis of five developmental milestones—walk, sit, first word, roll, and crawl acquisition. We compared the average age, in months, that each genotype met these milestones to the average age demonstrated by the DDST-II. The percent delays were then calculated for each genotype. While all genotypes were delayed in walking, individuals with variants in *ARID1A* were the most significantly delayed (64%). Individuals with variants in *SMARCB1* were most delayed in acquiring their first word (78%), rolling (57%), and sitting (55%). While all genotypes were delayed in crawling, *ARID1A* individuals (44%) were the most significantly delayed. Oddly enough, individuals with variants in *SOX11* were early (50%) in meeting their roll milestone. Individuals with variants in *SOX11* also met their first word (44%), sit (10%), and crawl (10%), milestones earlier than other genotypes.

#### 3.1.5. Quantitative Traits

To more rigorously assess differences in quantitative traits across genotype groups, we compared mean values in age at developmental milestone acquisition and birth length/weight (Figure 1B–D). For roll acquisition, there was a general trend towards patients with SMARCB1 variants having delayed acquisition of this developmental milestone compared to all other gene groups, with significant differences specifically seen between patients with ARID1B variants and those with SMARCB1 variants (*p* = 4.0 × 10^−2^). Similarly, there was a general trend towards patients with SMARCB1 variants having delayed sit acquisition compared to all other genotype groups, with significant differences seen between SMARCB1 patients and ARID1B patients (*p* = 6.0 × 10^−4^), and ARID1A patients (*p* = 3.0 × 10^−2^) and SMARCA4 patients (*p* = 5.0 × 10^−2^). Lastly, patients with ARID2 pathogenic variants were found to have significantly shorter birth lengths compared to almost all other genotype groups (ARID2 vs. ARID1B *p* = 4.8 × 10^−9^, ARID2 vs. SMARCA4 *p* = 1.9 × 10^−8^, ARID2 vs. ARID1A *p* = 4.3 × 10^−7^, ARID2 vs. SMARCB1 *p* = 5.1 × 10^−7^, ARID2 vs. SOX11 *p* = 1.5 × 10^−3^).

### 3.2. Figures, Tables and Schemes

Figure 1A visualizes 29 classic qualitative phenotypes presented in individuals with variants in the BAF complex. We ran a by-gene analysis to assess the differences in the frequency of these phenotypes across genotypes. The results of this analysis are presented in a bar graph. The bars represent the frequency of each phenotype in each genotype. *ARID*-related variants are indicated on the bar graph in shades of blue, *SMARC*-related variants are presented in shades of pink, and *SOX11* is presented in yellow. Numbers above each bar represent the *p*-values for phenotypes with nominally statistically significant differences.

Figure 1B–D visualizes seven classic quantitative phenotypes presented in individuals with variants in the BAF complex. We ran a by-gene analysis to assess the differences in the frequency of these phenotypes across genotypes. The results of this analysis are presented in bar graphs. The bars represent the frequency of each phenotype in each genotype. *ARID*-related variants are indicated on the bar graph in shades of blue, *SMARC*-related variants are presented in shades of pink, and *SOX11* is presented in yellow. Numbers above each bar represent the *p*-values for phenotypes with nominally statistically significant differences. Vertical lines above the bars represent the margin of error for each phenotype.

**Table 1 genes-12-00937-t001:** CSS/BAF-related disorders registry population.

Gene	Individuals in the Sample Population
*ARID2*	8 (4%)
*ARID1A*	15 (7%)
*ARID1B*	130 (63%)
*SMARCA4*	32 (15%)
*SMARCB1*	14 (7%)
*SMARCE1*	4 (2%)
*SOX11*	5 (2%)
Total	208

**Table 2 genes-12-00937-t002:** Classic phenotype expression in individuals with variants in the BAF complex.

Genes Sample Population	*ARID1A* (*n* = 15)	*ARID1B* (*n* = 130)	*ARID2* (*n* = 8)	*SMARCA4* (*n* = 32)	*SMARCB1* (*n* = 14)	*SMARCE1* (*n* = 4)	*SOX11* (*n* = 5)	Total Cohort (*n* = 208)
**Features**	%	%	%	%	%	%	%	%
Hypertrichosis	5/15 **33%**	81/130 **62%**	1/8 **13%**	14/32 **44%**	7/14 **50%**	0/4 **0%**	1/5 **20%**	109/208 **52%**
Sparse Scalp Hair	8/15 **53%**	62/130 **48%**	5/8 **63%**	10/32 **31%**	7/14 **50%**	3/4 **75%**	3/5 **60%**	98/208 **47%**
Hypoplasia of the Distal Phalanx	8/15 **53%**	43/130 **33%**	1/8 **13%**	20/32 **63%**	6/14 **43%**	4/4 **100%**	3/5 **60%**	85/208 **41%**
Microcephaly	2/15 **13%**	11/130 **8%**	1/8 **13%**	12/32 **38%**	4/14 **29%**	0/4 **0%**	1/5 **20%**	31/208 **15%**
Macrocephaly	1/15 **7%**	31/130 **24%**	1/8 **13%**	1/32 **3%**	3/14 **21%**	0/4 **0%**	0/5 **0%**	37/208 **18%**
**Genitourinary Complications**								
Cryptochidism (Males)	3/9 **33%**	36/82 **44%**	0/8 **0%**	2/11 **18%**	3/6 **50%**	0/4 **0%**	1/5 **20%**	45/125 **36%**
Kidney Anomalies	5/15 **33%**	8/130 **6%**	0/8 **0%**	5/32 **16%**	2/14 **14%**	1/4 **25%**	0/5 **0%**	21/208 **10%**
Gastrointestinal Complications								
GERD	3/15 **20%**	22/130 **17%**	2/8 **25%**	10/32 **31%**	4/14 **29%**	2/4 **50%**	0/5 **0%**	43/208 **21%**
Chronic Constipation	3/15 **20%**	27/130 **21%**	2/8 **25%**	6/32 **19%**	2/14 **14%**	0/4 **0%**	1/5 **20%**	41/208 **20%**
**Cardiac**								
Atrial Septal Defect	3/15 **20%**	15/130 **12%**	0/8 **0%**	1/32 **3%**	2/14 **14%**	2/4 **50%**	0/5 **0%**	23/208 **11%**
**Neurological**								
Hypotonia	7/15 **47%**	57/130 **44%**	3/8 **38%**	12/32 **38%**	5/14 **36%**	3/4 **75%**	2/5 **40%**	89/208 **43%**
Seizures	3/15 **20%**	17/130 **13%**	0/8 **0%**	1/32 **3%**	2/14 **14%**	1/4 **25%**	1/5 **20%**	25/208 **12%**
Global Developmental Delays	7/15 **47%**	46/130 **35%**	4/8 **50%**	13/32 **41%**	6/14 **43%**	2/4 **50%**	2/5 **40%**	80/208 **38%**
Agenesis of the Corpus Callosum	6/15 **40%**	31/130 **24%**	1/8 **13%**	8/32 **25%**	5/14 **36%**	1/4 **25%**	0/5 **0%**	52/208 **25%**
Non-Verbal	3/15 **20%**	9/130 **7%**	0/8 **0%**	3/32 **9%**	1/14 **7%**	0/4 **0%**	0/5 **0%**	16/208 **8%**
ASD	2/15 **13%**	21/130 **16%**	0/8 **0%**	0/32 **0%**	2/14 **14%**	0/4 **0%**	0/5 **0%**	25/208 **12%**
Intellectual Disability	5/15 **33%**	42/130 **32%**	1/8 **13%**	4/32 **13%**	3/14 **21%**	1/4 **25%**	1/5 **20%**	57/208 **27%**
**Spine**								
Scoliosis	4/15 **27%**	19/130 **15%**	2/8 **25%**	4/32 **13%**	4/14 **29%**	1/4 **25%**	1/5 **20%**	35/208 **17%**
Opthamologic								
**Ptosis**	5/15 **33%**	17/130 **13%**	0/8 **0%**	5/32 **16%**	1/14 **7%**	0/4 **0%**	1/5 **20%**	29/208 **14%**
Strabismus	8/15 **53%**	40/130 **31%**	1/8 **13%**	10/32 **31%**	7/14 **50%**	0/4 **0%**	1/5 **20%**	67/208 **32%**
**Respiratory**								
Obstructive Sleep Apnea	1/15 **7%**	14/130 **11%**	0/8 **0%**	5/32 **16%**	1/14 **7%**	0/4 **0%**	0/5 **0%**	21/208 **10%**
Tracheomalacia	1/15 **7%**	21/130 **16%**	1/8 **13%**	3/32 **9%**	3/14 **21%**	0/4 **0%**	0/5 **0%**	29/208 **14%**
Laryngomalacia	3/15 **20%**	14/130 **11%**	1/8 **13%**	4/32 **13%**	2/14 **14%**	0/4 **0%**	0/5 **0%**	24/208 **12%**
**Hearing Loss**								
Sensorineural	2/15 **13%**	10/130 **8%**	1/8 **13%**	4/32 **13%**	2/14 **14%**	1/4 **25%**	0/5 **0%**	20/208 **10%**
Conductive	3/15 **20%**	9/130 **7%**	0/8 **0%**	4/32 **13%**	1/14 **7%**	1/4 **25%**	0/5 **0%**	18/208 **9%**
Unspecified	1/15 **7%**	14/130 **11%**	0/8 **0%**	3/32 **9%**	1/14 **7%**	0/4 **0%**	1/5 **20%**	20/208 **10%**
Hearing Loss Total	6/15 **40%**	33/130 **25%**	1/8 **13%**	11/32 **34%**	4/14 **29%**	2/4 **50%**	1/5 **20%**	58/208 **28%**
**Quantitative Measurements**								
Birth Weight (kg)	2.94 (15%ile) −1.06 SD	3.38 (39%ile) −0.29 SD	2.89 (14%ile) −1.07 SD	2.98 (15%ile) −0.93 SD	2.67 (6%ile) −1.43 SD	2.15 (1%ile) −2.26 SD	2.28 (2%ile) −2.05 SD	2.76 (10%ile) −1.28 SD
Birth Length (cm)	49.37 (38%ile) −0.30 SD	48.55 (27%ile) −0.60 SD	32.99 (<1%ile) −6.41 SD	49.35 (38%ile) −0.30 SD	49.87 (46%ile) −0.11 SD	45.72 (5%ile) −1.66 SD	48.01 (21%ile) −0.80 SD	46.26 (7%ile) −1.46 SD
**Developmental Milestones**	Age (% delay)	Age (% delay)	Age (% delay)	Age (% delay)	Age (% delay)	Age (% delay)	Age (% delay)	Age (% delay)
Age Aquired Roll (months)	9 (33%)	8 (25%)	8 (25%)	8 (25%)	14 (57%)	9 (33%)	4 (-50%)	9 (33%)
Age Acquired Sit (months)	12 (25%)	11 (18%)	11 (18%)	13 (31%)	20 (55%)	11 (18%)	10 (10%)	13 (31%)
Age Acquired Crawl (months)	16 (44%)	14 (36%)	12 (25%)	14 (36%)	15 (40%)	14 (36%)	10 (10%)	14 (36%)
Age Acquired First Word (months)	30 (63%	26 (58%)	17 (35%)	25 (52%)	41 (73%)	31 (65%)	16 (31%)	27 (55%)
Age Acquired Walk (months)	33 (64%)	23 (48%)	24 (50%)	32 (63%)	22 (21%)	27 (27%)	23 (48%)	26 (54%)

**Figure 1 genes-12-00937-f001:**
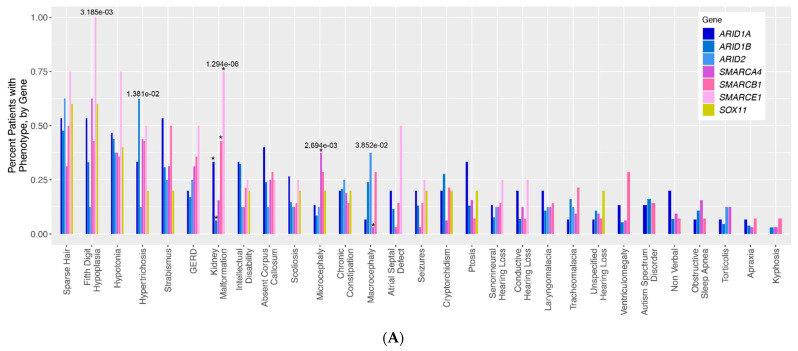
(**A**). Qualitative phenotypes in individuals with variants in the BAF complex. (**B**–**D**). Quantitative phenotypes in individuals with variants in the BAF complex.

## 4. Discussion

The purpose of this study was to examine genotype-phenotype correlations in a large cohort of individuals from the CSS/BAF-related disorders registry in an effort to identify possible differences across genotype groups. Overall, we found that the phenotypes classically described in CSS, including sparse scalp hair, fifth digit hypoplasia, hypotonia, and hypertrichosis, remained amongst the most common phenotypes reported for all CSS genotype groups. Furthermore, despite previous implications that *SMARC* and *ARID*-related variants display some degree of phenotypic differentiation, our data revealed largely similar phenotypes across all groups. Only eight phenotypes were identified with nominally statistically significant differences across CSS genotype groups: fifth digit hypoplasia, hypertrichosis, kidney malformations, microcephaly, macrocephaly, age at roll acquisition, age at sit acquisition, and birth length. Of these, only kidney malformations, age at sit acquisition, and birth length can truly be said to be significantly different between genotype groups after correcting for multiple testing. Consistent with what has been previously published, kidney malformations were found to be more common in individuals with *SMARCB1* and *SMARCE1* variants and underrepresented in individuals with *ARID1B* variants. Similarly, consistent with previous publications, individuals with *SMARCB1* variants were found to have significantly greater developmental delays compared to other groups. Interestingly, the significantly shorter birth length in individuals with *ARID2* variants is a finding that has not been previously reported.

There are a number of caveats to our study, with perhaps the greatest being the relatively small number of patients upon which we can base our conclusions. While large numbers of patients with *ARID1B* (*n* = 130) and *SMARCA4* (*n* = 32) were included in our cohort, we could identify only small numbers of individuals with variants in any of the other known CSS genes. Although phenotype correlations were observed within each cohort, it is difficult to make accurate conclusions regarding phenotypic differences in gene groups with limited data, and this must be kept in mind when considering the results of our statistical analyses. Another caveat of our study is its dependence on parent-reported phenotypes and on assessments by different medical providers without a single standard approach to phenotypic description. Individuals’ phenotypes are initially inputted by caregivers but ultimately verified with available medical records. While this approach allowed us to ascertain a greater number of CSS patients for inclusion, it also introduces potential bias and noise in the reporting of phenotypes across patients.

## 5. Conclusions

This manuscript demonstrates that, while there are some genotype-phenotype correlations in variants in the BAF complex, they are not as strong as previously hypothesized. Patients with CSS/BAF-related disorders should still get broad-spectrum genetic testing to distinctly define their BAF-related disorder. The increased risk of neurodevelopmental and organ-related complications should be considered in all individuals with variants in the BAF complex. To prevent and maintain the symptoms of CSS/BAF-related disorders, patients should utilize occupational, physical, and speech therapies. Patients may also benefit from specialty care, such as gastrointestinal, neurology, ophthalmology, nephrology, cardiology, and audiology specialists. The expansion of modern genetic testing and contemporary research may assist families in mitigating these challenges.

## Data Availability

The data presented in this study are available on request from the corresponding author. The data are not publicly available due to IRB restrictions.

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
