# Peer review of "Genotype-Phenotype Correlations in 208 Individuals with Coffin-Siris Syndrome"

_genes, 2021, doi:10.3390/genes12060937_

Round 1
Reviewer 1 Report
The manuscript is interesting and it is clear and well writing. The series contains enough cases with molecular studies to get conclussions.
Minor issue is that there are some typos such as includingwalk
Author Response
Reviewer 1
The manuscript is interesting and it is clear and well writing. The series contains enough cases with molecular studies to get conclusions.
Thank you for this positive feedback.
Minor issue is that there are some typos such as includingwalk
We have re-reviewed the manuscript for typos, thank you.
Reviewer 2 Report
The article presents an interesting, well documented genotype-phenotype analysis of a large cohort of patients affected by Coffin-Siris syndrome. In comparison to what already available in literature it provides a broader view, considering a large number of phenotypes and a great number of gene variations.
Introduction
It could be of benefit for readers if Authors could add more information on the syndrome main clinical manifestations.
Please add ORPHAcode to the OMIM code.
Materials and Methods
The quality of the paper would benefit by sharing more information on the registry and the cohort described.
Concerning the registry: starting date, users, governance, tools used for data collection, geographical coverage, data quality control in place, etc.
Phenotypes frequency in the Results would be more consistent if calculated only within a single mutation.
Please specify how were the 5 development milestones picked? They are more focused on motor skills, leaving out cognitive skills which are important when evaluating child development.
Data Quality: while the selected cohort is large, as the authors themselves state, some of the variations considered are represented in very limited numbers e.g. SMARCE1 (4 cases); SOX11 (5 cases), thus statistical significance is debatable.
Results
Please add information on the cohort: distribution per age, location (country coverage).
There’s an issue with Figure 2 which is not fully readable.
Discussion
Please revise the part stating the limits of the study highlighting the issues above described.
There is an issue with the data collection of phenotypic features. Are they reported by parents or by clinicians or both? The data quality of this information would greatly benefit from the use of HPO terms. Please refer to this issue in the discussion.
Author Response
Reviewer 2
It could be of benefit for readers if Authors could add more information on the syndrome main clinical manifestations.
We have added an additional sentence to the end of the first paragraph detailing some more common manifestations of CSS.
Please add ORPHAcode to the OMIM code.
Orphacode ORPHA:1465 has been added.
Materials and Methods
The quality of the paper would benefit by sharing more information on the registry and the cohort described.
We have added some additional information under Materials and Metholds regarding the registry.
Concerning the registry: starting date, users, governance, tools used for data collection, geographical coverage, data quality control in place, etc.
We have added additional information on this as well.
Phenotypes frequency in the Results would be more consistent if calculated only within a single mutation.
The data displayed in figure 1 shows the frequency of each phenotype within each gene group. That is, for example, 75% of patients with a SMARCE1 variant were found to have sparse hair, and thus are indicated in this way on the plot. The order that the phenotypes are presented in, along the horizontal axis of the plot, is in descending order of frequency of the phenotype for the entire CSS cohort, not dividing by gene. We can further clarify or modify the figure if this explanation does not address the reviewer’s concern.
Please specify how were the 5 development milestones picked? They are more focused on motor skills, leaving out cognitive skills which are important when evaluating child development.
Because cognitive skill assessments were not available for most of the individuals (or they varied greatly in the assessment tools) we decided to pick 5 significant milestones that were most prevalent among the cohort examined.
Data Quality: while the selected cohort is large, as the authors themselves state, some of the variations considered are represented in very limited numbers e.g. SMARCE1 (4 cases); SOX11 (5 cases), thus statistical significance is debatable.
We agree that it is difficult to draw meaningful conclusions about statistically-significant differences between groups when some groups have very little data. We have been careful to highlight these caveats in the manuscript, and have now expanded on the discussion of these caveats in the text of the result and discussion (page 4 line 121 and page 8, line 227). Unfortunately, we are limited by the number of patients in our cohort with variants in these two genes, and must do the best we can with the data that we have.
Results
Please add information on the cohort: distribution per age, location (country coverage).
We have added additional information available of the distribution of the registry. We did not go into great detail as we did not want this to detract from the purpose of the paper.
There’s an issue with Figure 2 which is not fully readable.
We have reinserted a new Figure 2 which we hope is more readable.
Discussion
Please revise the part stating the limits of the study highlighting the issues above described.
There is an issue with the data collection of phenotypic features. Are they reported by parents or by clinicians or both? The data quality of this information would greatly benefit from the use of HPO terms. Please refer to this issue in the discussion.
We had addressed this with the below text but added the underlined sentence for clarification.
‘Another caveat of our study is its dependence on parent-reported phenotypes and on assessments by different medical providers without a single standard approach to phenotypic description. Individuals’ phenotypes are initially inputted by caregivers but ultimately verified with available medical records. While this approach allowed us to ascertain a greater number of CSS patients for inclusion, it also introduces potential bias and noise in the reporting of phenotypes across patients.’
Reviewer 3 Report
Review-2021-Genes-1250029-Coffin Siris syndrome-090621
Genotype-phenotype correlations in individuals with Coffin-Siris syndrome
By
Vasko A et al
The paper presents clinical features organized in a structured fashion and based on molecularly confirmed cases based on 341 participants in a registry on CSS/BAF group of patients. Among them 208 patients had sufficient phenotypic data and were all molecularly confirmed with the majority being affected by pathogenic variants in two genes:ARID1B (130) and SMARCA4 (32), comprising 78% of all included patients.
The structured clinical presentation is valid and noteworthy even if it is an issue for concern in terms of consistency that the scores on clinical features have been reported by many different parents and many different doctors/caregivers, apparently.
The presentation is clear and with valuable figures and the main findings include that the clinical presentation of patients with pathogenic variants in BAF complex is broader in terms of organ affection and developmental/intellectual delay and that patients therefore must be followed by a true multidiscliplinary team with access to all relevant specialists.
The study does not reveal great surprises in terms of what is previously known , but illustrates how a cohort of patients with variants in the same complex of genes can be registered and followed and the summarized natural history can be broadened, especially if the ways of collecting the clinical information could be more systematic and stringent.
Author Response
Thank you for your kind comments and thorough review of the manuscript. We understand that while this report does not necessarily offer great new insight, we are pleased that it gets the point across that these individuals do need extensive and thorough multidisciplinary care.